# Delayed Colorectal Cancer Diagnosis during the COVID-19 Pandemic in Alberta: A Framework for Analyzing Barriers to Diagnosis and Generating Evidence to Support Health System Changes Aimed at Reducing Time to Diagnosis

**DOI:** 10.3390/ijerph18179098

**Published:** 2021-08-28

**Authors:** Emily Walker, Yunting Fu, Daniel C. Sadowski, Douglas Stewart, Patricia Tang, Bethany Kaposhi, Heather Chappell, Paula Robson, Sander Veldhuyzen van Zanten

**Affiliations:** 1Surveillance and Reporting, Advanced Analytics, Cancer Research & Analytics, Cancer Care Alberta, Alberta Health Services, Edmonton, AB T5J 3H1, Canada; yuntingfu@hotmail.com (Y.F.); bethany.kaposhi@albertahealthservices.ca (B.K.); 2Division of Gastroenterology, Department of Medicine, University of Alberta, Edmonton, AB T6G 2X8, Canada; sadowski@ualberta.ca (D.C.S.); vanzanten@ualberta.ca (S.V.v.Z.); 3Cancer Strategic Clinical Network, Alberta Health Services, Edmonton, AB T5J 3H1, Canada; Douglas.Stewart@albertahealthservices.ca; 4Department of Oncology, Cumming School of Medicine, University of Calgary, Calgary, AB T2N 4N2, Canada; Patricia.Tang@albertahealthservices.ca; 5Cancer Research & Analytics, Cancer Care Alberta, Alberta Health Services, Edmonton, AB T5J 3C6, Canada; heather.chappell@albertahealthservices.ca (H.C.); paula.robson@albertahealthservices.ca (P.R.); 6Digestive Health Strategic Clinical Network, Alberta Health Services, Edmonton, AB T5J 3E4, Canada

**Keywords:** colorectal cancer, COVID-19, delayed diagnosis, colonoscopy capacity, health system planning, administrative data analysis

## Abstract

The frequency of colorectal cancer (CRC) diagnosis has decreased due to the COVID-19 pandemic. Health system planning is needed to address the backlog of undiagnosed patients. We developed a framework for analyzing barriers to diagnosis and estimating patient volumes under different system relaunch scenarios. This retrospective study included CRC cases from the Alberta Cancer Registry for the pre-pandemic (1 January 2016–4 March 2020) and intra-pandemic (5 March 2020–1 July 2020) periods. The data on all the diagnostic milestones in the year prior to a CRC diagnosis were obtained from administrative health data. The CRC diagnostic pathways were identified, and diagnostic intervals were measured. CRC diagnoses made during hospitalization were used as a proxy for severe disease at presentation. A modified Poisson regression analysis was used to estimate the adjusted relative risk (adjRR) and a 95% confidence interval (CI) for the effect of the pandemic on the risk of hospital-based diagnoses. During the study period, 8254 Albertans were diagnosed with CRC. During the pandemic, diagnosis through asymptomatic screening decreased by 6·5%. The adjRR for hospital-based diagnoses intra-COVID-19 vs. pre-COVID-19 was 1.24 (95% CI: 1.03, 1.49). Colonoscopies were identified as the main bottleneck for CRC diagnoses. To clear the backlog before progression is expected, high-risk subgroups should be targeted to double the colonoscopy yield for CRC diagnosis, along with the need for a 140% increase in monthly colonoscopy volumes for a period of 3 months. Given the substantial health system changes required, it is unlikely that a surge in CRC cases will be diagnosed over the coming months. Administrators in Alberta are using these findings to reduce wait times for CRC diagnoses and monitor progression.

## 1. Introduction

The first case of COVID-19 was reported in Alberta, Canada on 5 March 2020. From that time, in common with other Canadian and international jurisdictions, various health system and public health measures were put in place to limit the spread of infection and maintain sufficient healthcare capacity to manage the pandemic [1]. In Alberta, these measures included stopping population-based cancer screening from April to June 2020, offering virtual care rather than in-person visits to cancer centres and primary care settings, decreasing access to diagnostic imaging and other diagnostic tests, and the implementation of a suite of public health orders and guidelines pertaining to physical distancing, gatherings, and travel [1]. While the provincial government did not introduce mandatory lockdowns during the pandemic, on 28 April 2020, Alberta Health updated preventative guidelines for COVID-19 to include a recommendation that Albertans stay home as much as possible and avoid non-essential travel [1]. This recommendation remained in place for the remainder of 2020. Concurrent with the instigation of these and other measures in Alberta, we observed a decline in the numbers of cancer cases being reported to the Alberta Cancer Registry (ACR), which is legally mandated to record all reportable cases of cancer in the province when notified by a physician and/or laboratory. This decline in registrations started in early April 2020 and persisted through the first wave of the COVID-19 pandemic. Similar observations were reported in other jurisdictions, with authors interpreting these declines in cancer registration as evidence of delayed diagnoses [2,3,4,5,6,7,8,9,10,11].

This work was initiated during the first wave of the pandemic, with the goal to estimate the impact of health system decisions on CRC diagnosis rates and develop an efficient strategy for addressing the short-term implications of this phenomenon. The evidence generated through this work was intended to inform health system planning designed to reduce the backlog while ensuring sufficient capacity to handle a potential surge in diagnoses. This work was initiated with colorectal cancer (CRC), which in Alberta is the second and third most commonly diagnosed cancer in males and females, respectively [12]. The components of the multipronged pandemic response instigated by Alberta Health Services (AHS) that directly impacted CRC diagnosis included cessation of all non-urgent colonoscopies. By mid-March, screening with the fecal immunochemical test (FIT) was also paused to reduce the volume of patients coming through clinics/laboratories. Since the number of COVID-19 cases was low and manageable for hospitals from March to May 2020, colonoscopies and FITs were partially resumed in mid-May after 10 and 9 weeks of total suspension, respectively. In July 2020, endoscopy units were given the approval to resume activities to 90% of their pre-COVID capacity. However, due to capacity constraints related to physical distancing, increased cleaning requirements, and staffing complications, most units were only able to achieve approximately 80% of their pre-COVID capacity at that time.

Because of ongoing capacity limitations throughout the first wave of the pandemic, we expected a growing backlog of patients awaiting a diagnosis of CRC. To address this, we had the following specific aims:(a)Characterize the impact of health system measures used to contain the COVID-19 pandemic on the number of CRC diagnoses, and changes in the diagnostic pathways and intervals relative to pre-pandemic patterns;(b)Monitor CRC progression in the Alberta populations, defined by diagnosis pre- or intra-pandemic;(c)Quantify the impacts of changes in the health system capacity or approaches to triaging patients for diagnostic procedures on the size of the backlog in CRC cases.

## 2. Materials and Methods

This study followed a retrospective cohort design. We identified two population-based cohorts for different aims of this work. The conceptual framework for this analysis was diagnostic pathways and intervals [13].

### 2.1. Diagnostic Pathway Framework

Diagnostic pathways are a sequence of events from initial clinical presentation to a final diagnosis of cancer [13]. These events include associated diagnoses, tests, and procedures undergone prior to receiving a cancer diagnosis. We refer to these events as diagnostic milestones. Diagnostic intervals are the length of time between initial clinical presentation and a final diagnosis, or between diagnostic milestones [13]. There are several advantages to applying the diagnostic pathway in this context. First, understanding the typical pathways through which patients are diagnosed with colorectal cancer highlights the impacts of public health and health system decisions that were intended to contain the pandemic, in terms of the creation of barriers or bottlenecks in the pathway and the subsets of the patient population most affected. Second, identification of barriers and bottlenecks in the pathway can be used to estimate the effects of altering capacity to remove those barriers on getting back to and exceeding expected patient volumes. Finally, diagnostic intervals place these events on an average timeline that can be used to predict the timing of diagnoses once public health and health system barriers are altered or eliminated.

This work was initiated through a review of the scientific literature and a clinical consultation. We searched MEDLINE and PubMed for scientific articles on diagnostic pathways and timelines for CRC. Additionally, the study team included multiple clinicians specializing in gastroenterology and colorectal tumours. The findings from the review and consultation were used to develop a preliminary diagnostic framework to guide the data abstraction and variable creation. A list of diagnostic milestones, including all relevant procedures, tests, and diagnoses that patients may experience prior to receiving a cancer diagnosis was generated and reviewed by clinical collaborators to ensure all relevant milestones were captured in the dataset. To increase the efficiency of measuring diagnostic pathways/intervals using administrative data, we developed a modified approach for defining the initial clinical presentation. Specifically, we used proxies for primary care encounters, such as lab tests ordered as part of initial clinical investigations. The preliminary diagnostic pathway developed through this phase of the project is shown in Figure 1. The estimates from an analysis of colorectal cancer diagnostic pathways using Alberta data were used as the expected distribution of patients across pathways [14].

### 2.2. Diagnostic Milestone Definitions

The data were obtained on all blood tests in the year prior to a colorectal cancer diagnosis measuring the following: iron, ferritin, mean corpuscular volume (MCV), and hemoglobin. The dates of all the FITs in the year prior to a colorectal cancer diagnosis were obtained. The following symptoms were included in the analysis: rectal bleeding (International Classification of Diseases version 10 (ICD-10) codes: K625, K552, K922); abdominal pain (ICD-10 codes: R100, R101, R102, R103, R104); unexplained weight loss (ICD-10 code: R634); changes in bowel habits (ICD-10 code: R194); and bowel obstructions (ICD-10 code: K566). The diagnostic procedures included: colonoscopies (Canadian Classification of Health Intervention (CCI) codes: 2NM70, 2NM71, 2NQ70, 2NQ71); colorectal surgeries (CCI codes: 1NQ87, 1NQ89, INM87, 1NM89, 1NM91); and abdominal computed tomography (CT) scans (CCI code: 3OT20).

### 2.3. Data Sources

The data for these analyses were obtained from multiple administrative and clinical datasets. The Alberta Cancer Registry (ACR) was used to identify all patients that had received a diagnosis of colorectal cancer during the study period. The data on relevant laboratory tests were obtained from Alberta Precision Labs. The Discharge Abstract Database (DAD) was used to identify all occurrences of relevant diagnoses, tests, and procedures occurring on an in-patient basis during the study period. The National Ambulatory Care Reporting System (NACRS) was used to identify all out-patient encounters related to colorectal cancer diagnoses during the study period.

### 2.4. Aim 1: Characterize the Impact of Health System Measures Used to Contain the COVID-19 Pandemic on the Number of CRC Diagnoses, and Changes in the Diagnostic Pathways and Intervals Relative to Pre-Pandemic Patterns

The Alberta Cancer Registry (ACR) was used to identify a cohort of patients aged 18 years and older who were diagnosed with colorectal cancer between 1 January 2016 and 1 July 2020. Among patients with multiple CRC diagnoses within the study period, only one diagnosis was randomly selected for inclusion in the analysis. The Personal Health Numbers (PHNs) and diagnosis dates were used to link the data from the ACR with the data on severe presentations of gastrointestinal disease, laboratory tests, and hospital-based diagnostic tests and procedures within the year prior to the diagnosis date. July 1 was selected as the cut off for this cohort because there was a 1–2-month lag in the data for the in-patient and out-patient diagnoses and procedures.

#### 2.4.1. Diagnostic Pathway Refinement

We developed a refined diagnostic pathway model to increase the amount of variation in the population that was captured by the framework. We followed an iterative process, guided by the initial clinically defined framework and clustering in the data. A numeric variable was created for each diagnostic milestone, with values representing the number of times each patient experienced that milestone within the year prior to their CRC diagnosis. A principal component analysis (PCA) was used to identify the diagnostic milestones that clustered together in the dataset, refining the initial diagnostic framework to reflect a greater amount of the variation in the population. The eigenvalues were set at 1 and factors were rotated using the promax method [10]. The linear combinations of diagnostic milestones identified through the PCA were used to generate a categorical variable, with each category reflecting different combinations of tests, diagnoses, and procedures completed as part of a colorectal cancer diagnosis. Patients that were not classifiable according to the refined diagnostic framework were examined and additional pathways were added to the variable definition as needed. Patients were excluded from the analysis if they had a cancer diagnosis within the study period, but no record of relevant diagnostic milestones, to avoid skewing interval estimates. To confirm that tests occurred in the expected chronological order, the number of days between diagnostic milestones was calculated. If the difference between the tests was a positive value, this was used as evidence that the hypothesized chronological order was correct.

#### 2.4.2. Interval Measurement

The number of days between each test was estimated. The total interval was estimated using the first identified test or symptom diagnosis related to colorectal cancer within the year prior to a colorectal cancer diagnosis identified in administrative datasets and the diagnosis date as recorded in the ACR. The number of days between the initial test in the dataset and the colorectal cancer diagnosis was estimated as the total diagnostic interval.

#### 2.4.3. Comparison of Diagnostic Pathways and Intervals Pre- and Post-COVID-19

All CRC diagnoses occurring between 1 January 2016 and 29 February 2020 were classified as pre-COVID-19 and all CRC diagnoses occurring between 1 March and 1 July 2020 were classified as intra-COVID-19. The difference in the proportion of pre- and intra-COVID-19 patients diagnosed through each pathway was estimated. The weighted average difference in the proportions of patients diagnosed through each pathway pre- and intra-COVID-19 was estimated for all the pathways initiated by the same tests or diagnoses using the number of patients diagnosed through each set of pathways pre-pandemic as weights. Decreases in the proportion of patients diagnosed through a given pathway or set of pathways were used as evidence that those subsets of patients were experiencing diagnostic delays due to the COVID-19 pandemic. The difference between median diagnostic interval lengths in days was estimated between patients diagnosed pre-pandemic relative to those diagnosed during the pandemic. The bootstrap method was used to calculate the 95% confidence intervals for median diagnostic interval days (with 5000 bootstrap replicates).

### 2.5. Aim 2: Monitor CRC Progression in the Alberta Populations, Defined by Diagnosis Pre- or Intra-Pandemic

A concern with a delayed diagnosis is the potential for stage shift, leading to a greater degree of morbidity and mortality and the need for increasingly aggressive medical interventions. Given that the staging data among recently diagnosed patients was not available, diagnosis while admitted to a hospital was used as a proxy for more severe disease at presentation. Patients were classified as being diagnosed while admitted to hospital if their diagnosis date coincided with a diagnostic test performed during a hospital admission, as identified in the DAD.

The proportion of patients diagnosed while admitted to a hospital was compared pre- and intra-COVID-19. The findings from Aim 1 highlighted that a certain proportion of patients were diagnosed with more severe disease, independent of the pandemic. The proportion of patients diagnosed through those pathways increased during the pandemic, indicating that factors leading to a diagnostic delay are more likely to impact patients with less severe disease. While logical, this finding complicates the assessment of the extent of disease progression in the patient population, as it highlights that comparisons of disease severity among patients diagnosed in the pre- and intra-pandemic periods are affected by selection bias. We used the modified Poisson regression to estimate the adjusted relative risks (RR) and 95% confidence intervals (CI) as measures of the effect of being diagnosed intra-pandemic versus pre-pandemic on a hospital-based diagnosis. The model covariates included age at CRC diagnosis, sex, and proxies for disease severity at initial presentation, including FIT results and diagnoses of bowel obstructions. The disease severity variables were included to adjust for the selection bias introduced by the differential impact of COVID-19-related diagnostic delays on patients with different degrees of disease severity [11].

### 2.6. Aim 3: Quantify the Impacts of Changes in the Health System Capacity or Approaches to Triaging Patients for Diagnostic Procedures on the Size of the Backlog in CRC Cases

A cohort of patients defined by having undergone any of the diagnoses, tests, or procedures identified a priori from January 2015 onward was identified from the DAD and the NACRS. The data were linked with the cohort of patients diagnosed with colorectal cancer from 1 January 2016 to 1 July 2020 using PHNs. January 2015 was selected as the start date for this cohort to obtain all the data on diagnostic tests in the year leading up to a 2016 diagnosis.

#### 2.6.1. Scenario Planning

The findings from Aim 1 were used to identify important parameters for scenario planning. Specifically, we examined diagnostic milestones in pathways for which there was a decrease in diagnoses intra-pandemic, with the assumption that the backlog is primarily comprised of these patients and bottlenecks in these pathways need to be addressed. Additionally, the diagnostic milestones commonest to the largest proportion of patients were analyzed. The procedure volumes from the same calendar months in 2019 were assumed to reflect the full capacity for the remaining months of 2020. A systematic review on timelines for CRC progression found that a colonoscopy occurring more than 9 months following a positive FIT was associated with higher odds of CRC and a diagnosis at advanced stages [12]. Therefore, the scenarios were defined by the capacity to meet or exceed pre-COVID-19 procedure volumes over the remaining months of 2020, as December 2020 marked 9 months from pausing screening and colonoscopies. Additionally, the variation in the proportion of procedures that yielded a colorectal cancer diagnosis was incorporated into each scenario.

#### 2.6.2. Statistical Analysis

The proportion of patients undergoing each diagnostic procedure that were diagnosed with colorectal cancer within the 30 days following the date of their test was estimated for each calendar month and year. The average colorectal cancer diagnosis yield by procedure, month, and year was estimated. The cumulative backlog of patients until the end of 2020 was estimated as the difference between expected CRC diagnoses and estimated number of diagnoses made possible under each scenario, defined by procedure capacity and percent yield of colorectal cancer diagnoses.

## 3. Results

There were 8254 patients diagnosed with CRC between 1 January 2016 and 1 July 2020 in Alberta. The demographic characteristics of these patients are shown in Table 1. From March to the end of August 2020, when this analysis was completed, the number of CRC diagnoses in Alberta declined by 55% when compared to the same months for the years 2016–2019. After June 2020, some diagnostic activities were resumed, which explained a small rebound in the number of CRC cases; however, the diagnoses did not return to expected volumes (644 average cases from 2016–2019; 20% difference). Overall, we estimated a backlog of approximately 467 undiagnosed CRC cases by the end of August 2020.

### 3.1. Aim 1: Characterize the Impact of Health System Measures Used to Contain the COVID-19 Pandemic on the Number of CRC Diagnoses, and Changes in the Diagnostic Pathways and Intervals Relative to Pre-Pandemic Patterns

#### 3.1.1. Diagnostic Pathways and Intervals

The distributions of diagnostic procedures are shown in Table 2. The refined diagnostic pathway model is shown in Figure 2 and explained 75.2% of the patients in the cohort. The diagnostic pathways fit into one of three categories that were initiated via: (1) screening, (2) blood tests, specifically those that a primary care physician would order to investigate gastrointestinal complaints or issues that may be related to CRC, and (3) symptoms, ranging from abdominal pain and rectal bleeding to bowel obstruction, diagnosed in an in-patient or ambulatory setting. The total interval durations for each diagnostic pathway are shown in Figure 3. The median number of days between each diagnostic milestone for all pathways are shown in Table 3.

#### 3.1.2. Comparison of Diagnostic Pathways and Intervals Pre- and Intra-COVID-19

There were 398 patients diagnosed with CRC intra-pandemic. Of these, 76.4% were classified into the identified diagnostic pathways. Pre-pandemic, the proportion of patients diagnosed through pathways initiated by screening, blood tests in primary care, or urgent care were 31·2%, 47·4%, and 20·5%, respectively. The proportion of patients diagnosed through the screening pathways was reduced by 6.5% during the pandemic (15.9% of patients diagnosed after March 2020). The proportion of patients diagnosed through pathways initiated by blood tests in primary care remained approximately the same pre- and intra-COVID-19 (pre-COVID-19, 47.4%; intra-COVID-19, 46.1%). Lastly, the proportion of patients diagnosed with symptomatic disease in urgent care increased by 2.7% during the pandemic, relative to before (38.2% of patients diagnosed after March 2020). The median number of days between each diagnostic milestone for all the pathways among patients diagnosed before the pandemic and during the pandemic are shown in Table 3.

### 3.2. Aim 2: Monitor CRC Progression in the Alberta Populations, Defined by Diagnosis Pre- or Intra-Pandemic

Overall, the proportion of patients diagnosed while admitted to hospital increased by 9% during the pandemic. Stratified by the diagnostic pathway, the increases were: screening, 1.0%; primary care, 9.2%; and urgent care, 5.6%. The unadjusted and adjusted RRs and 95% CIs for the effect of the pandemic period on a hospital diagnosis are shown in Table 4. There was a modest increased risk of an in-patient CRC diagnosis in the intra-pandemic period. However, the estimates should be interpreted with caution until a greater proportion of the backlog in CRC patients is cleared.

### 3.3. Aim 3: Quantify the Impacts of Changes in the Health System Capacity or Approaches to Triaging Patients for Diagnostic Procedures on the Size of the Backlog in CRC Cases

A total of 680,187 patients experienced ≥1 diagnostic milestones of interest between 1 January 2015 and 1 July 2020. Colonoscopies were the most common component in the diagnostic pathways (85.6% of patients). Therefore, the initial focus of the scenario planning was the throughput capacity and diagnostic efficiency of colonoscopies for identifying CRC cases. We made the following assumptions based on the findings from Aim 1: 85·6% of patients undergo at least one colonoscopy and no other bottleneck diagnostic tests/procedures. The analysis of the data from the pre-pandemic period showed that 1·66% of colonoscopies completed each month yielded a CRC diagnosis.

The analysis was completed in September 2020. The estimated backlog of CRC diagnoses over the remaining months of 2020 with different scenarios of colonoscopy capacity and different CRC diagnosis yields for completed scopes are shown in Figure 4. In order for the estimated backlog of CRC diagnoses to be cleared within 9 months of the start of the pandemic, the volume of colonoscopies and yield of CRC diagnoses needed to increase beyond 100% of the pre-COVID-19 levels. Specifically, increases of 140% in colonoscopy volume and 4% in CRC diagnosis yields each month would be needed for a period of 3 months. This scenario would correspond with an average estimated surge in CRC diagnoses beyond the normal case registration volumes of 96.7% per month from October to December 2020.

## 4. Discussion

The findings of decreased cancer diagnoses in Alberta during the pandemic are consistent with those from other jurisdictions within Canada and internationally [2,3,4,5,6,7,8,9]. Additionally, evidence of the impact of cessation of screening has impacted cancer diagnosis rates is consistent with the other findings in the literature [2,3,4,5,6,7,8,9]. The diagnostic pathways analysis yielded evidence of variation in how patients were diagnosed with CRC in the pre- versus intra-pandemic periods and in the average diagnostic intervals. We were able to identify which subsets of the patient population were most affected by intra-COVID-19 health system changes. We found that asymptomatic screening was reduced during the pandemic relative to pre-pandemic rates. These findings are consistent with what is known about the contextual factors influencing diagnosis in Alberta during the first six months of the COVID-19 pandemic: specifically, the cessation of screening and reduction of in-person visits with primary care physicians where asymptomatic screening may be initiated were influential. In addition, a follow-up colonoscopy for patients with positive FIT results was curtailed due to the closure of endoscopy units. This reduction in the colonoscopy capacity was seen in many other healthcare systems worldwide [20,21]. Our findings suggest a combination of approaches would be needed to clear the colonoscopy backlog. Various strategies have been suggested, including a reduction of colonoscopy for average risk screening and enhanced utilization of non-invasive fecal tests [20]. Our analysis suggests that the capacity to complete colonoscopies would need to increase beyond 100% of the pre-pandemic monthly volumes, with >2x the yield of CRC diagnoses from completed colonoscopies. Achieving these targets requires identifying strategies for removing the constraints around colonoscopy capacity post-pandemic. Prioritization for colonoscopies, specifically improving triage for patients with a higher risk of having CRC based on a combination of clinical and demographic characteristics, would align the colonoscopy resources with patients most likely to benefit [21]. There is increasing recognition that a significant number of surveillance colonoscopies for post-polypectomy follow-up are performed in patients who are at lower risk for colorectal cancer, particularly those with 1–2 low risk adenomas [13,14,15]. Returning these patients to average risk screening with the FIT rather than a colonoscopy would significantly reduce the colonoscopy burden [16].

Given the challenges around achieving the identified targets for both of these parameters in a short time frame, it was not possible to clear the estimated backlog in patients by the end of the 9-month period from the start of the pandemic. However, evidence from this analysis was used to advocate for maintenance of the services throughout the remainder of the pandemic and into the post-pandemic period. Because of this, further dips in CRC registrations were not seen to the same magnitude throughout the remainder of 2020 and into early 2021. Since no catch-up period was observed, there remains a cohort of undiagnosed patients from the start of the COVID-19 pandemic. The framework for the analysis presented in this paper will be used for ongoing monitoring of CRC diagnosis and for scenario planning after the pandemic subsides.

Targeting the most common diagnostic procedures is the most efficient strategy for eliminating the backlog of patients. The current analysis focused on the colonoscopy, a test undergone by the largest proportion of patients, occurring early in the diagnostic pathway. Additionally, the timelines were assessed by month. However, this analysis could be expanded to incorporate capacity and yield adjustments to other bottlenecks later in the diagnostic pathways and a more refined timeline, based on the average diagnostic intervals as estimated in this analysis. Specifically, the next steps would be to include cancer-suspected surgeries, which 54.3% of patients in this analysis underwent prior to a CRC diagnosis. An ongoing analysis of months with higher versus lower yields of CRC diagnoses by procedure is being used to identify patient characteristics that could inform triage guidelines going forward. Future analyses will investigate the effect of subsequent waves of the COVID-19 pandemic on CRC diagnosis in Alberta.

There are several limitations to these analyses. First, primary care encounters beyond the lab tests were not included in the dataset. Therefore, symptoms identified through primary care leading to referrals for diagnostic tests were missed and not included in the diagnostic interval measurements. Second, the changes in healthcare-seeking behaviours during the pandemic were beyond the scope of this study but may impact diagnostic intervals. Finally, the estimated effect of the pandemic on in-patient CRC diagnoses may be due in part to the residual effects of selection bias and should be interpreted with caution at this point in time.

## 5. Conclusions

We developed a comprehensive framework for generating evidence that could support health system approaches for reducing the backlog of undiagnosed CRC patients and planning for subsequent surges in diagnosis. Health systems will need to address how best to rapidly increase their colonoscopy volume for several months if they are to resolve the backlog of undiagnosed CRC patients. Given the substantial changes required in both the procedure capacity and patient triaging to increase the CRC diagnosis yield from colonoscopies, it is unlikely that a surge in CRC diagnoses will occur over the coming months. The evidence from this work has been shared with key stakeholders in Alberta’s health system to ensure its use in decision making around the capacity to complete procedures and approaches to triaging patients through the system. In addition to the immediate benefits of this work for operational planning during the pandemic, this work has potential utility in operational planning long term to achieve improvements in the accessibility and efficiency of clinical operations.

## Figures and Tables

**Figure 1 ijerph-18-09098-f001:**
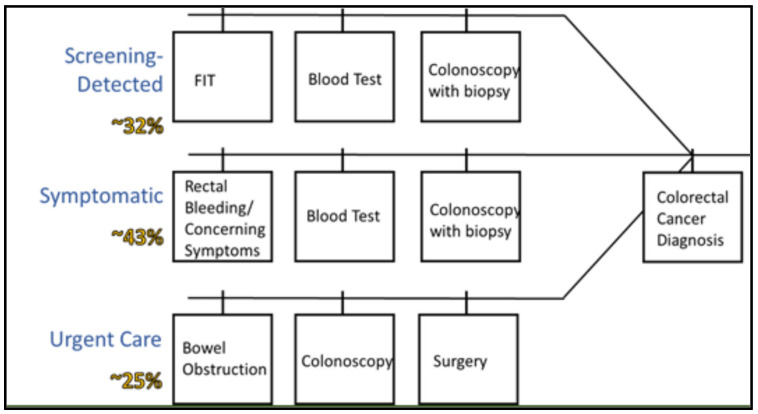
Preliminary diagnostic framework developed to guide data abstraction, with estimated distribution of patients across diagnostic pathways from the literature [14,15,16,17,18,19].

**Figure 2 ijerph-18-09098-f002:**
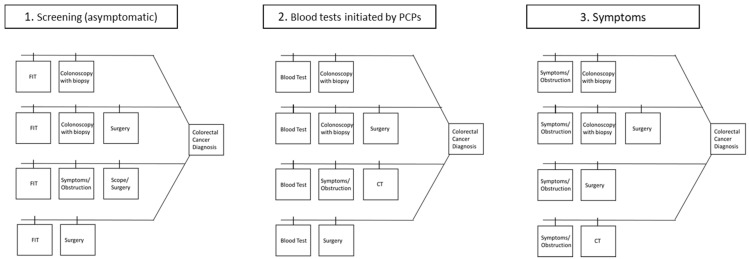
Refined diagnostic pathway frameworks among 7545 patients diagnosed with colorectal cancer between 1 January 2016 and 1 July 2020 in Alberta, Canada. PCP = Primary care physicians.

**Figure 3 ijerph-18-09098-f003:**
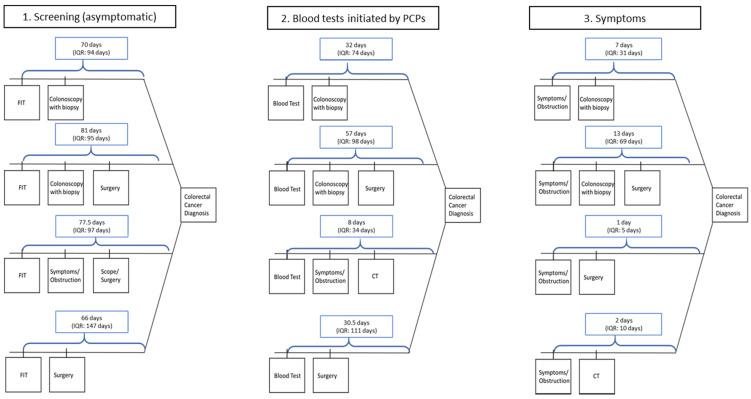
Median diagnostic intervals in days and interquartile range (IQR) by pathway among 7241 patients diagnosed with colorectal cancer between 1 January 2016 and 1 March 2020. PCP = Primary care physicians.

**Figure 4 ijerph-18-09098-f004:**
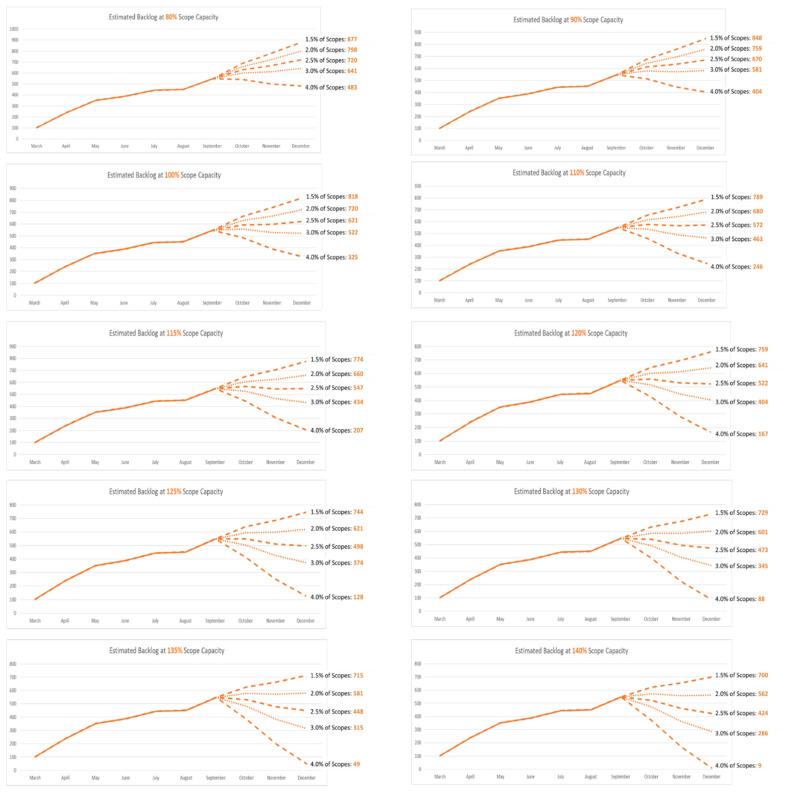
Estimated backlog in CRC diagnoses over the remaining months of 2020 (September–December) under different scenarios defined by monthly colonoscopy capacity and colorectal cancer diagnosis yield.

**Table 1 ijerph-18-09098-t001:** Frequency and distribution of patient characteristics and tests/procedures within 1 year prior to diagnosis among 7545 patients diagnosed with colorectal cancer between 1 January 2016 and 1 July 2020 in Alberta, Canada.

Patient Characteristics	Total		Diagnosed Pre-COVID		Diagnosed Intra-COVID	
	N	%	N	%	N	%
Age at Diagnosis						
18–39 years	262	3.47	240	3.31	22	7.24
40–49 years	541	7.17	513	7.08	28	9.21
50–59 years	1402	18.58	1354	18.70	48	15.79
60–69 years	2043	27.08	1971	27.22	72	23.68
70–79 years	1850	24.52	1779	24.57	71	23.36
≥80 years	1447	19.18	1384	19.11	63	20.72
Sex						
Male	4325	57.32	4162	57.48	163	53.62
Female	3219	42.66	3078	42.51	141	46.38
Cancer Site						
Colon	5070	67.20	4862	67.15	208	68.42
Rectosigmoid Junction	303	4.02	290	4.00	13	4.28
Rectum	2172	28.79	2089	28.85	83	27.30
Diagnosed while Admitted to Hospital						
No	5901	78.21	5689	78.57	212	69.74
Yes	1644	21.79	1552	21.43	92	30.26

**Table 2 ijerph-18-09098-t002:** Frequency and distribution of diagnostic milestones among 7545 patients diagnosed with colorectal cancer in Alberta, Canada between 1 January 2016 and 1 July 2020.

Diagnostic Test/Procedure	Total		Diagnosed Pre-COVID-19		Diagnosed Intra-COVID-19	
	N	%	N	%	N	%
FIT						
No	5617	74.45	5349	73.87	268	88.16
Yes	1928	25.55	1892	26.13	36	11.84
Blood Test						
0	2392	31.72	2261	31.22	132	43.42
1	4320	57.26	4165	57.52	155	50.99
≥2	832	11.03	815	11.26	17	5.59
Colonoscopy						
0	1090	14.45	1026	14.17	64	21.05
1	5979	79.24	5763	79.59	216	71.05
≥2	476	6.31	452	6.24	24	7.90
Surgery						
0	3451	45.74	3305	45.64	146	48.03
1	3756	49.78	3609	49.84	147	48.36
≥2	338	4.48	327	4.52	11	3.62
CT Scan						
0	5366	71.12	5152	71.15	214	70.39
1	1925	25.51	1846	25.49	79	25.99
≥2	254	3.37	243	3.36	11	3.62
Bowel Obstruction						
0	6850	90.79	6579	90.86	271	89.14
1	535	7.09	511	7.06	24	7.89
≥2	160	2.12	151	2.09	9	2.96
Rectal Bleed						
0	6688	88.64	6426	88.74	262	86.18
≥1	857	11.36	815	11.26	42	13.82
Abdominal Pain						
0	6972	92.41	6694	92.45	278	91.45
≥1	573	7.59	547	7.55	26	8.55

**Table 3 ijerph-18-09098-t003:** Median intervals, interquartile ranges (IQR), and 95% confidence intervals (CI) in days between diagnostic milestones among 7545 patients diagnosed with colorectal cancer between 1 January 2016 and 1 July 2020 in Alberta, Canada.

Diagnostic Interval Boundaries	Diagnosed Pre-COVID			Diagnosed Intra-COVID		
	Median (Days)	IQR	95% CI	Median (Days)	IQR	95% CI
Intervals Initiated by Screening						
FIT → Colonoscopy	72	86	(69, 76)	108.5	92	(61, 129.5)
FIT → Surgery	76	90	(71, 79)	117	92	(56.5, 134)
FIT → CT	49	106.5	(36, 57.5)	-	-	-
FIT → Symptoms ^§^	58	98	(44, 66)	118	223	(21, 257)
Intervals Initiated by Blood Tests						
Blood Test → Colonoscopy	55	95	(52, 56)	42	116	(25, 58)
Blood Test → Surgery	60	99	(57, 62)	41	115.5	(19, 56)
Blood Test → CT	17	75	(14, 21)	0	5.5	-
Blood Test → Symptoms ^§^	24	84	(16, 31.5)	10	178	(0, 165)
Intervals Initiated by Symptoms ^§^/Bowel Obstructions						
Symptoms → Scope	3	31	(2, 3)	2·5	15	(1, 5)
Symptoms → Surgery	3	32	(2, 3)	5	18	(1, 11)
Symptoms → CT	0	3	-	0	1	-
Intervals Between Diagnostic Tests and Final Diagnosis Dates						
Colonoscopy → Diagnosis	0	0	-	0	0	-
Surgery → Diagnosis	0	0	-	0	0	-
CT → Diagnosis	7	44	(5, 7)	10	61	(4, 12)

^§^ Symptoms and bowel obstructions are those diagnosed in an in-patient or ambulatory setting, including rectal bleeding, abdominal pain, and unexplained weight loss.

**Table 4 ijerph-18-09098-t004:** Unadjusted and adjusted relative risks (RR) and 95% confidence intervals (CI) for the effects of patient characteristics and diagnosis period for being diagnosed while admitted to hospital.

Patient Characteristics	Unadjusted		Adjusted	
	RR	95% CI	RR	95% CI
Age at Diagnosis				
1 Year Increase	1.02	1.02, 1.03	1.01	1.01, 1.02
Sex				
Male	Ref		Ref	
Female	1.13	1.03, 1.23	1.05	0.96, 1.14
FIT Testing				
No	Ref		Ref	
Yes	0.37	0.32, 0.43	0.47	0.41, 0.54
Bowel Obstructions				
No	Ref		Ref	
Yes	3.46	3.19, 3.75	2.88	2.64, 3.15
Pandemic Period				
Pre-COVID-19	Ref		Ref	
Intra-COVID-19	1.41	1.16, 1.72	1.24	1.03, 1.49

## Data Availability

The data from these analyses were obtained from multiple administrative datasets containing data on cancer diagnoses and healthcare encounters in Alberta. The email address for cancer data requests in Alberta is ACB.cancerdata@albertahealthservices.ca.

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
