# Peer review of "Delayed Colorectal Cancer Diagnosis during the COVID-19 Pandemic in Alberta: A Framework for Analyzing Barriers to Diagnosis and Generating Evidence to Support Health System Changes Aimed at Reducing Time to Diagnosis"

_ijerph, 2021, doi:10.3390/ijerph18179098_

Round 1

Reviewer 1 Report

Thank you for inviting me to review the manuscript entitled ``Delayed Colorectal Cancer Diagnosis during the COVID-19 Pandemic in Alberta: a framework for analyzing barriers to diagnosis and generating evidence to support health system changes aimed at reducing time to diagnosis´ (ID ijerph-1302885) submitted by Walker et al. 

This study focuses on evaluating the diagnosis of colorectal cancers and its implications during the first five months (Mar - July, 2020) in the  COVID-19 pandemic, and also comparing with previous diagnosis in the pre-pandemic period.

There is a lack of information in regards to the measures established by the local governments to mitigate COVID-19 transmission, i. e. Had people been instructed to isolate themselves indoors or to maintain social isolation and physical distancing? Was there a regional lockdown implementation? After that, have people returned to their “normal life”? In addition, authors may provide data after the intra-pandemic period. How many wavers has the territory experienced? This should impact the diagnosis of colorectal cancers. 

It seems that only one article was used to choose the variables of the study. 

  1. Sikdar, K.C.; Dickinson, J.; Winget, M. Factors Associated with Mode of Colorectal Cancer Detection and Time to Diagnosis: A 428 Population Level Study. BMC Health Services Research 2017, 17, 7, doi:10.1186/s12913-016-1944-y. Based on that, the authors reported the following statement: “This work was initiated through review of the scientific literature and clinical consultation. Findings from the review and consultation were used to develop a preliminary diagnostic framework to guide data abstraction and variable creation.” In this regard,  which types of articles were used to review the processes applied for diagnosing colorectal cancer? Meta-Analysis, Systematic-Reviews, Original articles, Clinical Trials, Case reports? There is a lack of information and references. 

The authors evaluated data from Mar-July, 2020. But they are showing data until Aug, 2020: “From March-August 2020, the number of CRC diagnoses in Alberta declined by 55% when compared to the same months for the years 2016-2019. From June-August 2020, some diagnostic activities were resumed which explained a small rebound in the number of CRC.”

Improving the quality of the pictures will be of interest to the reader.

The major drawback is not to show data after the intra-pandemic period. These data may open new avenues if “normal life” can facilitate diagnosis of colorectal cancer. I encourage the authors in demonstrating and comparing those variables.

Author Response

Reviewer Comment:

There is a lack of information in regards to the measures established by the local governments to mitigate COVID-19 transmission, i. e. Had people been instructed to isolate themselves indoors or to maintain social isolation and physical distancing? Was there a regional lockdown implementation? After that, have people returned to their “normal life”? In addition, authors may provide data after the intra-pandemic period. How many wavers has the territory experienced? This should impact the diagnosis of colorectal cancers. 

Response:

We have added more detail in the introduction on measures intended to mitigate transmission of COVID-19 that indirectly and directly impacted CRC diagnosis.

Reviewer Comment:

It seems that only one article was used to choose the variables of the study. 

  1. Sikdar, K.C.; Dickinson, J.; Winget, M. Factors Associated with Mode of Colorectal Cancer Detection and Time to Diagnosis: A 428 Population Level Study. BMC Health Services Research 2017, 17, 7, doi:10.1186/s12913-016-1944-y. Based on that, the authors reported the following statement: “This work was initiated through review of the scientific literature and clinical consultation. Findings from the review and consultation were used to develop a preliminary diagnostic framework to guide data abstraction and variable creation.” In this regard,  which types of articles were used to review the processes applied for diagnosing colorectal cancer? Meta-Analysis, Systematic-Reviews, Original articles, Clinical Trials, Case reports? There is a lack of information and references. 

Response:

There were multiple articles used to select identify diagnostic milestones, along with clinical consultation with gastroenterologists and oncologists specializing in colorectal tumours. The paper cited here is the paper that contained expected distributions of the patients across pathway types. We have clarified in the text the review methods, the clinical specialties involved in identifying variables for the analysis and the included citations for additional literature reviewed.

Reviewer Comment:

The authors evaluated data from Mar-July, 2020. But they are showing data until Aug, 2020: “From March-August 2020, the number of CRC diagnoses in Alberta declined by 55% when compared to the same months for the years 2016-2019. From June-August 2020, some diagnostic activities were resumed which explained a small rebound in the number of CRC.”

Response:

The analysis aiming to estimate the capacity scenarios that would lead to clearing the backlog in patients was completed in September, so we used all data available to us at that time to estimate the size of the backlog. The pathways analysis was completed to inform those scenarios, but data delays meant that we had to cut the cohort at July 1. We have added sentences clarifying why different dates were used.

Reviewer Comment:

Improving the quality of the pictures will be of interest to the reader.

Response:

We have provided higher resolution images.

Reviewer Comment:

The major drawback is not to show data after the intra-pandemic period. These data may open new avenues if “normal life” can facilitate diagnosis of colorectal cancer. I encourage the authors in demonstrating and comparing those variables.

Response:

We plan to conduct follow-up analyses as more long-term data on the entire pandemic becomes available, including factors like stage at diagnosis. This is outside of the scope of the original work, which was conducted at the start of the pandemic in order to plan for potential case volumes and eliminate a backlog from the initial pandemic period within the period of time before progression is expected. We have included more information about next steps for this analysis in the conclusion.

Reviewer 2 Report

The diagnosis of colorectal cancer (CRC) has decreased during the pandemic in Alberta, Canada. A considerable backlog and health-system planning are necessary to address that issue and identify and resolve the bottleneck. This is a retrospective study with an objective to characterize the impact of COVID-19 pandemic on the number of CRC diagnoses and in the diagnostic pathways and intervals relative to pre-pandemic patterns. The authors also monitored the CRC progression in the population of Alberta populations pre and intra-pandemic

The authors also aimed to estimate the impacts of the pandemic on the health system capacity.

This paper has several limitations.

1) This paper is focused on the logistics of healthcare management and might have some value to the policymakers.

2) Introduction is very short and did not make enough argument about the necessity of the study. It is not clear to me whether similar studies have been done in the past, not necessarily limited to Alberta or CRC. One line in the intro may not be enough. The study design is not clear to me.

3) I can not comment on the word count, but it looks like more than 2000 words have been used only to describe the methods. It is lengthy, lacks focus, and is hard to follow. At times it read like a paper on logistics drifting from the main focus.

4) The authors described the diagnostic pathway framework. How did they build the framework? Is this a new innovation or based on previously described works? I do not see any reference whatsoever.

5) Section 2.3. Diagnostic Milestone Definitions: this section was introduced out of context and abruptly. The readers would be lost. Where from did it come, why introduced, and where would it lead us to…

6) Likewise, the very next paragraph.. ‘2.4. Diagnostic Pathway Refinement’ fell short as the goal and purpose of refining the diagnostic pathway were unclear.

7) While reading the results, I see that the authors described, “The estimated backlog of CRC diagnoses over the remaining months of 2020 with different scenarios of colonoscopy capacity and different CRC diagnosis yields for completed scopes are shown in figure 4.” However, figure 4 is illegible; thus not possible for the reviewers to understand the implication of the quoted result.

8) I understand from this paper that the diagnosis of CRC could vary depending on the presentation, settings, and clinical scenario, and the pandemic could limit lots of preventive workup in asymptomatic patients, thus limiting the ability to diagnose asymptomatic patients. However, to my understanding, this paper only gathered some data that compared pre vs. intra-pandemic scenarios. However, this paper fell short due to its lack of clarity in presentation, unnecessarily prolonged methods, and descriptive results with unclear significance.

9) I do not see any comparison with previously published results.

10) Abstract: consider rewriting the sentence “Modified Poisson regression analysis was used to estimate the adjusted relative risk (adjRR) and 95% confidence interval (CI) for the effect of the pandemic on the risk of hospital

based diagnoses.

Author Response

Reviewer Comment:

1) This paper is focused on the logistics of healthcare management and might have some value to the policymakers.

Response:

This is correct, we are unsure why this is listed as a limitation. The purpose of this work is to support health system decision makers in clearing the backlog of patients and planning for surges in patients and changing patient needs as a result of delayed diagnoses.

Reviewer Comment:

2) Introduction is very short and did not make enough argument about the necessity of the study. It is not clear to me whether similar studies have been done in the past, not necessarily limited to Alberta or CRC. One line in the intro may not be enough. The study design is not clear to me.

Response:

We have added more detail in the introduction about the rationale for the work. The study design is described in the methods

Reviewer Comment:

3) I can not comment on the word count, but it looks like more than 2000 words have been used only to describe the methods. It is lengthy, lacks focus, and is hard to follow. At times it read like a paper on logistics drifting from the main focus.

Response:

This is contrary to most other reviews we have received on this paper, which is to ensure the methods are as detailed as possible. We conducted multiple complex analyses for this work and think that the detail in the methods is appropriate. We have organized the methods by analytic aim, so that it is clear to the reader what the methods were intended to achieve. We have corrected the numbering issue to clarify which methods are intended to achieve which analytic aim.

Reviewer Comment:

4) The authors described the diagnostic pathway framework. How did they build the framework? Is this a new innovation or based on previously described works? I do not see any reference whatsoever.

Response:

We have added a reference to diagnostic pathways for cancer and described how we built our pathway in detail in the methods.

Reviewer Comment:

5) Section 2.3. Diagnostic Milestone Definitions: this section was introduced out of context and abruptly. The readers would be lost. Where from did it come, why introduced, and where would it lead us to…

Response:

This section comes directly after a paragraph defining the term “diagnostic milestones” and explaining how we decided which milestones to target in the data. This section provides the codes used to search for these tests/procedures/diagnoses in the data so that readers will know exactly what was included in our dataset and the analysis can be replicated. We do not think this section is abrupt or out of context.

Reviewer Comment:

6) Likewise, the very next paragraph.. ‘2.4. Diagnostic Pathway Refinement’ fell short as the goal and purpose of refining the diagnostic pathway were unclear.

Response:

We added a sentence to explain the purpose of refining the initially defined framework.

Reviewer Comment:

7) While reading the results, I see that the authors described, “The estimated backlog of CRC diagnoses over the remaining months of 2020 with different scenarios of colonoscopy capacity and different CRC diagnosis yields for completed scopes are shown in figure 4.” However, figure 4 is illegible; thus not possible for the reviewers to understand the implication of the quoted result.

Response:

We have provided higher resolution images.

Reviewer Comment:

8) I understand from this paper that the diagnosis of CRC could vary depending on the presentation, settings, and clinical scenario, and the pandemic could limit lots of preventive workup in asymptomatic patients, thus limiting the ability to diagnose asymptomatic patients. However, to my understanding, this paper only gathered some data that compared pre vs. intra-pandemic scenarios. However, this paper fell short due to its lack of clarity in presentation, unnecessarily prolonged methods, and descriptive results with unclear significance.

Response:

We have added more to the introduction to clarify the rationale for the work. We think that the length of the methods is appropriate given the amount of work we completed. The results include both descriptive and adjusted estimates. We have added some statements to the conclusions about the significance of this work.

Reviewer Comment:

9) I do not see any comparison with previously published results.

Response:

We have added some comparison with other analyses in the discussion.

Reviewer Comment:

10) Abstract: consider rewriting the sentence “Modified Poisson regression analysis was used to estimate the adjusted relative risk (adjRR) and 95% confidence interval (CI) for the effect of the pandemic on the risk of hospital based diagnoses.

Response:

We think that this sentence is a clear description of the methods used in the analysis and do not think that it needs to be re-written.

Reviewer 3 Report

This paper addresses an important public health crisis in ongoing Covid -19 pandemic. The analysis presented here for health system measures needed to address delayed diagnosis of CRC in covid times are valuable and can also be applied to other diseases which may need early diagnosis and interventions. Article is nicely written and authors have described in details their aims, methodology and results to justify the significance of the study. I recommend this study to be published in IJERPH.

Author Response

Thank you for your feedback.

Round 2

Reviewer 2 Report

Agree with the revised manuscript.